# The Use of Physical Restraints on Geriatric Patients: Culture and Attitudes among Healthcare Professionals at Intermediate Care Hospitals in Majorca. A Qualitative Study Protocol

**DOI:** 10.3390/ijerph18147509

**Published:** 2021-07-14

**Authors:** Alba Carrero-Planells, Ana Urrutia-Beaskoa, Cristina Moreno-Mulet

**Affiliations:** 1Department of Nursing and Physiotherapy, University of the Balearic Islands, 07122 Palma, Spain; cristina.moreno@uib.es; 2Care, Chronicity, and Health Evidences Research Group, University of the Balearic Islands, 07122 Palma, Spain; 3Fundación Cuidados Dignos, 48300 Gernika, Spain; anaurrutia@cuidadosdignos.org; 4Care, Chronicity, and Health Evidences Research Group, Health Research Institute of the Balearic Islands (IdISBa), 07010 Palma, Spain

**Keywords:** physical restraint, health knowledge, attitudes, practice, organisational culture, health personnel, safety

## Abstract

The use of physical restraints is a common practice in the care of hospitalised and institutionalised elderly people. This use is determined by factors related to the patients, their families, the healthcare professionals, the institution, and prevailing social values. Today, however, this practice is often questioned because of its physical, psychological, moral, ethical, and legal repercussions. The present study explores attitudes among healthcare professionals towards the physical restraint of geriatric patients in intermediate care hospitals in Majorca. This study is based on a qualitative design, combining an ethnomethodological approach with critical discourse analysis. The theoretical framework is drawn from Foucault’s work in this field and from Haslam’s theory of mechanisation. Individual interviews will be conducted with physicians, nurses, and nursing assistants at intermediate care hospitals in Majorca. The analysis will focus on these professionals’ knowledge, attitudes, and practices regarding the use of such measures, seeking to identify the factors, especially institutional factors, that determine the use of restraints. It is essential to determine the prevailing culture among healthcare professionals regarding the use of physical restraints on geriatric patients in order to design and propose a more dignified health care model in which such restraints are eliminated.

## 1. Introduction

Population ageing is associated with a greater prevalence of cognitive and functional impairment, which increases the number of dependent older people who must be cared for in hospitals and residential centres [1]. These people are sometimes disoriented, agitated, or wandering, and may present postural alterations and/or an increased risk of falls. In addition, they may have attached medical devices. These numerous healthcare challenges are often managed by means of physical restraint systems [2]. An international consensus statement defines physical restraints as “any action or procedure that prevents a person’s free body movement to a position of choice and/or normal access to his/her body by the use of any method, attached or adjacent to a person’s body that he/she cannot control or remove easily” [3]. The use of physical restraint is considered a form of involuntary treatment, together with the use of psychotropic medication and non-consensual care [4].

The use of restraints is common, known to exist in acute, chronic, and residential settings, and is closely related to the characteristics of the institutions and the profile of their users [5]. The prevalence of physical restraints in nursing homes and long-term care centres varies considerably among different countries, from an average of 6% in Switzerland to over 31% in Canada [6]. In Spain, the prevalence ranges from 23.8% to 36% [7,8] but rises to 86% when the use of bed rails is included [8]. A study conducted in eight European countries in nursing home residents suffering from dementia revealed a prevalence of 31.4%, ranging from 6.1% in France to 83.2% in Spain [9]. The most common means of physical restraint are bilateral bed rails, trunk or limb belts, and devices such as mobility-limiting chairs or immobilising sheets [1,2,10,11,12,13,14,15].

In most cases, restraints are used as safety devices [1,14,16]. Healthcare professionals justify their use as necessary to prevent falls, control agitation or wandering, achieve postural control, and facilitate the application of therapeutic measures or procedures. In addition, they are sometimes employed in response to requests for support by families [1,2,10,11,13,17]. However, on many occasions, the indication for this type of measure is related to achieving organisational goals, maintaining a comfortable social environment, alleviating problems caused by insufficient or inadequately trained care staff, or correcting defects in the design of physical spaces [12,13].

Residents’ characteristics are decisive factors in the use of physical restraints. Cognitive and physical impairments, disorientation, severe morbidity, and low scores on indexes of activities of daily living are all important risk factors for the use of restraints [18]. However, this situation is influenced not only by individual conditions but also by pressure from relatives; the knowledge, opinions, and preferences of healthcare professionals; normal practices of the institution; and social values [2,5,10,19]. In Spain, there is no national normative or legislative framework regarding this question [20]. On an international level, several documents must be considered. The Universal Declaration of Human Rights and the European Convention of Human Rights state that everyone has the right to life, liberty, and security of person [21,22,23]. The United Nations Principles for older persons dictates that older persons should be able to enjoy human rights and fundamental freedoms when residing in any shelter, care, or treatment facility, including full respect for their dignity, beliefs, needs, and privacy and for the right to make decisions about their care and the quality of their lives [24]. In some countries, such as Denmark, United Kingdom, Germany, the Netherlands, Australia, and the United States, physical restraint use is legislated. These measures seem to have contributed to a reduction in restraint use in these countries [11,25].

The knowledge, attitudes, and practices of healthcare professionals jointly shape the “philosophy of care” regarding the use of restraints. In this respect, Fariña evaluated levels of knowledge among healthcare professionals about physical restraints, concluding that although in general the professionals’ knowledge in this area could be considered acceptable, there were some misconceptions regarding the available alternatives, the handling of behavioural alterations, and the possible relationship between restraint and death [2]. In general, healthcare professionals’ attitudes towards physical restraint were ambiguous, and they conflicted between respect for the residents’ dignity and autonomy on one hand and the need to ensure their safety on the other [2,16,26]. Specifically, nursing staff described the burdens of responsibility related to accident or injury, expressing concerns about families’ complaints and possible litigation following accidents [16]. Many nurses value physical restraints as a necessary measure to prevent falls and ensure geriatric patients’ health and well-being, thus legitimising their use [13,26]. However, when physical restraints are used against the patient’s will, nurses may experience a moral dilemma, resulting in feelings of frustration, discomfort, and concern [26]. To address this ethical conflict, nurses may rationalise the use of restraints by viewing them as measures to protect the patient’s safety [19,27,28]. This attitude has been described as moral blindness [29]. Nurses may also share negative feelings in this respect with their colleagues [19,27] or consider alternative measures [27]. There is an evident association between healthcare professionals’ knowledge, attitudes, and practices and the use of physical restraints. Thus, the greater the knowledge about the use of restraints, the less likely they are to be used in healthcare practice [2].

Relevant institutional factors in this context include staff shortages that may be experienced and the inability to provide geriatric patients with appropriate supervision. The use of physical restraint systems may be spurred by a perceived lack of alternative measures and/or problems arising from the architectural characteristics of the centre [5,13,16,19,28,30,31]. Finally, the absence of quality assurance systems for work routines, practices, and designs, accompanied by poor management, may result in patients’ needs being viewed as secondary to those of the staff, which could foster the use of restraint mechanisms [30].

Another important factor is that some family members present ambivalent attitudes towards the use of physical restraint or are openly opposed to this practice, viewing it as degrading and humiliating. Others, however, consider these measures to be appropriate and positive, approving of them as safety elements, viewing them as helpful in preventing falls, and relying on professional criteria [26]. Families’ lack of knowledge in this field has been identified as a determining factor leading to greater use of restraints [14,32]. Another study concluded that family members, and even users, participate in decision-making in this area and initiate the process [10]. Family members’ requests are based on worries for patients’ safety [5,10].

The use of physical restraint in care settings has been questioned for various reasons, one of which is the risk of physical and/or psychological complications. Physical complications include pressure ulcers, loss of muscle tone, and the risk of suffocation. Among the possible psychological complications are agitation, fear, and feelings of humiliation [33]. Furthermore, physical restraints have been represented as an attack on basic human rights, such as freedom, dignity, and autonomy [11,19,34,35,36]. Gastmans analysed the ethical values and norms related to the use of physical restraints. The application of physical restraint often corresponds with a disproportionate infringement of the principle of respect for the autonomy of older people. The ability of human beings to make choices must always be respected in this context. Furthermore, when considering older people as full persons, we must accept their wellbeing involves more than just preventing physical harm. Finally, in view of supporting geriatric patients’ self-reliance optimally, the application of physical restraint methods should be considered only in exceptional circumstances, when a serious risk is posed to older people or others, or if attempts to avoid physical restraint are unsuccessful [34]. In addition, their regular use is considered an indicator of low quality of care and may even reflect physical or psychological abuse. For all these reasons, an absence of physical restrictions, whenever possible, is recommended as the standard of care for geriatric patients [11]. Accordingly, various policies have been implemented to reduce the use of such restraints [11,15,25,37,38,39,40,41], including educational programmes and consultation or guidance by an expert nurse [15]. Indeed, studies have shown that the removal of physical restraints does not increase the number of falls, the severity of associated injuries [11,15,25,37,38], or the use of psychotropic drugs [11,25,37,38,39,40]. In consequence, these restraints are neither necessary nor effective elements in preventing falls and injuries in individuals with behavioural problems [11,37,41]. Nevertheless, they are still a common practice in the care of geriatric patients. In recent years, influential calls have been made to promote restraint-free care. For example, the Spanish Society of Geriatrics and Gerontology published a consensus document recommending that mechanical and pharmacological restraints should be considered an exceptional resource to be used in a timely, rational, and proportional manner and only after the failure of all other measures available to control the situation [17].

In short, healthcare without physical restraint should be the paradigm of quality geriatric care. However, to achieve this, a better understanding is needed of the knowledge, attitudes, and practices regarding physical restraint in intermediate care [15]. Only by understanding the reasons for its use in this context will it be possible to determine and address the underlying factors. In particular, further qualitative research is needed to identify the most significant aspects of restraint from the standpoint of the healthcare professionals involved [2]. These study aims arise from widespread concerns about this practice due to the perceived lack of respect for the dignity of those subjected to physical restraint, and they are expected to provide a reasoned basis for the future elimination of this practice. 

### 1.1. Theoretical Framework

This study can be framed within the critical social paradigm. Specifically, the ideas of Michel Foucault provide the basis for our analysis of the practices of professionals and the institutional culture of the physical restraint of geriatric patients. In this respect, the conceptual framework related to safety and discipline is of particular importance [42,43]. As a second-level approach, theories related to dehumanisation are also taken into consideration. In the latter respect, dehumanisation is defined as not perceiving other individuals as entirely human with full rights and obligations [44]. In consequence, they are located beyond the margins within which moral and social norms apply, making them more vulnerable to potentially degrading treatment. Specifically, our analysis refers to Haslam’s theory of mechanisation [45], which is applied to situations in which individuals are seen as lacking warmth, emotion, and individuality, and thus are compared to inanimate objects. Mechanisation can be observed in the fields of technology and medicine, whereby patients are viewed as machines, some of whose components need to be repaired [44].

### 1.2. Study Aims

The main aim of this study is to enhance our understanding of the culture of physical restraint of geriatric patients by referring to the attitudes of healthcare professionals (physicians, nurses, and nursing assistants) at intermediate care hospitals in Majorca (Spain). Specifically, we examine the knowledge, attitudes, and practices of these professionals related to the use of physical restraints and describe the institutional factors that influence this question, identifying the similarities and differences in the discourse of the healthcare professionals according to their area and level of work.

## 2. Methods and Analysis

### 2.1. Research Design, Study Setting, Participants, and Recruitment

This protocol describes a qualitative study with an ethnomethodological design, including critical discourse analysis. Ethnomethodology is dedicated to explicating the ways in which members collectivity create and maintain a sense of order and intelligibility in their social life. The focus in ethnomethodological studies is always on procedural aspects of participants’ situated practices. One of the most important elements of this design is the investigators’ reflexivity process [46]. From an ethnomethodological standpoint, the social world’s facticity is accomplished by way of members’ interactional work, the mechanics of which produce and maintain the accountable circumstances of their lives [47]. This social inquiry describes the actions taken by people in a given context—in the present case, the routine use of physical restraints in intermediate care hospitals—and interprets the meanings attributed to this behaviour. A significant characteristic of this methodology is its holistic, contextual nature, which facilitates the analysis of professionals’ discourse in their institutional context [48]. 

The study participants will be physicians, nurses (both those directly providing care and those with management responsibilities), and nursing assistants—these professionals are identified as the main ones involved in decision making, prescription, and application of physical restraints—working in three intermediate care hospitals in Majorca. The Joan March Hospital and the General Hospital are second-level public hospitals, dependent on and indirectly managed by a referral hospital. The third, the Sant Joan de Déu Hospital in Palma (SJD Palma Hospital), is privately managed, but through an agreement of association with the Balearic Islands Health Service, it forms part of the Public Hospital Network.

The study population is made up of approximately 45 physicians, 150 nurses, and 180 nursing assistants who are employed in the three hospitals included in the study. A theoretical–intentional sampling will be carried out [49], which enables the inclusion of participants with the greatest narrative and communicative capacity to address the research question. At the same time, this sampling method provides a comprehensive view of the phenomenon under study by including all the professional profiles involved and illustrating their different perspectives on the question.

Regarding the number of participants in the study, at least seven healthcare professionals will be recruited from each job category, along with four individuals in management positions, or as many as necessary to achieve data saturation. The profiles of these participants will be segmented by gender and workplace, taking into account the number of workers in each category (Table 1). Table 2 shows the inclusion and exclusion criteria that will be applied to the centres/units and the healthcare professionals considered. Discursive differences will be analysed according to the professional category. According to the critical social paradigm, we assume that physical restraint is wielded as a practice of power over geriatric patients and is related to the hierarchical position of the professional who prescribes or applies this restraint.

Formal and informal participant recruitments are currently being employed in the workplace. Firstly, the study was presented to the respective hospital managers and their permission and collaboration was requested. These managers were also informed of the need for a collaborator, or key informant, in each hospital to facilitate recruitment. These individuals—two nursing coordinators of hospitalization units and a research manager—are now making the research project more widely known, explaining its nature and purpose to the professionals who meet the criteria for inclusion. Those who express interest will be sent an email with detailed information, including the study purpose, data collection techniques, benefits and risks derived from participation in the study, ethical issues related to confidentiality and volunteering, and contact details of the principal investigator, and will be invited to participate. In addition, recruitment is occasionally conducted by the snowball strategy—from healthcare professionals who have participated in the study, other professionals with discourses for and against the use of physical restraints are contacted.

### 2.2. Data Compilation

Data for this study will be obtained via semi-structured individual interviews [50]. The questions posed in these interviews (see Table 3) are aimed at determining the opinions and experiences of (non-management) healthcare professionals regarding their use of physical restraints. The responses will be analysed and used to design subsequent interviews with hospital managers. At the beginning of each interview, the participant will be invited to complete a form providing the following sociodemographic information: age, sex, area of work, years of professional experience, and specific training in the use of physical restraints. Throughout the study, a field diary will be completed by the main researcher and, optionally, by the participants.

This study has a scheduled duration of three years, starting with data collection in 2020 and 2021. In the first phase, pilot interviews were carried out with a healthcare professional from each area of work. In the second phase, interviews are currently being held with healthcare professionals. Finally, in the third phase, the managers will be interviewed, and follow-up interviews with the healthcare professionals will also take place if necessary.

### 2.3. Results Analysis

A content and critical discourse analysis will be carried out. Critical discourse analysis studies the ways in which social power abuse, dominance, and inequality are enacted, reproduced, and resisted, focusing on both text and speech communications in the social and political contexts. This type of analysis seeks to understand, expose, and ultimately overcome social inequality [51]. 

The results obtained will be coded and categorised by inductive–deductive analysis (abductive analysis). The data analysis will be systematised and optimised with the MAXQDA 2020 computer program. 

### 2.4. Scientific Rigour

To ensure methodological rigour and validity, information will be collected until data saturation is reached. Moreover, investigator, techniques, and theoretical framework triangulation will be carried out. Finally, the reflexivity of the principal investigator will be examined throughout the process, especially during the data collection and analysis phase, through the researcher’s field diary.

### 2.5. Ethical Considerations

The study was designed in accordance with the provisions of the Declaration of Helsinki and as approved by the Nursing Directorate and the Research Committee at each of the hospitals included. In addition, the project was favourably evaluated by the Research Ethics Committee of the Balearic Islands (IB 4026/19 PI, December 2019). The voluntary nature of participation will be respected, and the confidentiality of the data will be ensured, in accordance with current rules and regulations. All participants will be sent a detailed information sheet and asked to sign the informed consent document prior to the interview. The latter document clearly states that participation in the study is voluntary and that this consent may be revoked at any time.

## 3. Discussion

The use of physical restraints for geriatric patients as part of the care process is a problem that must be addressed in order to optimise care and to improve patients’ quality of life. The transition from a “culture of restraint” to a “culture of no restraint” requires time, awareness, sensitivity, and adaptation to internalise and assimilate such a change in the care paradigm [17]. The aim of this study is to address the question from the perspective of the healthcare professionals involved, seeking to understand why physical restraints for geriatric patients are still used and to identify the elements that may impede their withdrawal. In Spain, prior studies in this respect include that of Fariña [2], who quantitatively examined the knowledge, attitudes, and practices of nurses and nursing assistants in nursing homes in relation to the use of physical restraints. Among the conclusions drawn, this study highlighted the need to explore the professionals’ perspectives from a qualitative approach, thus reflecting the opinions gathered in greater depth. In accordance with the latter view, the present study uses a qualitative methodology to obtain detailed knowledge about the beliefs, prejudices, and practices of healthcare professionals; the dynamics within staff groups; and the influence of social values regarding the use of physical restraints in the care of geriatric patients.

Some research in this area of study has been conducted in nursing homes [2,5,10,13], but little or none has been conducted with respect to geriatric intermediate care units. Nevertheless, this care environment is one of growing importance for patients with chronic conditions, facilitating their admission to hospital units for convalescence and rehabilitation. Accordingly, the main aim of the present study is to identify and analyse the institutional factors inherent to intermediate care that influence the use of physical restraints, a question that has been identified in previous research as a crucial aspect of care practice [1,30]. Furthermore, it is essential to contextualise the discourse of the professionals concerned in order to propose plans for the removal of physical restraints, in accordance with the circumstances of each care centre and hospital unit.

Among the outcomes expected from this study, we hope to identify the factors that determine the use of restraints as it concerns the patients themselves, their families, the healthcare professionals, and the institution. Such factors may include the educational training received [2,15], beliefs about restraints such as protectors [2,16], the evaluation and understanding of patients’ behaviour, and the professional’s responsibility towards the care recipient [28]. Furthermore, the discourse is expected to vary according to the respondent’s area of professional activity. Particular attention will be paid to the concept of safety, in terms of the patient’s physical safety, the conditions of the care environment and the legal safety of the professionals. In addition, a Foucaultian analysis of the concept of security will be carried out [42,43]. The study will inquire into professionals’ day-to-day experiences regarding the use of physical restraints in the care of geriatric patients, identifying possible ethical conflicts and the strategies adopted in response [26].

Regarding the relationship between knowledge, attitudes, and practices [2], we believe that healthcare professionals who generally approve of the use of physical restraints in the circumstances described may normalise the situation and make more extensive use of these measures. On the other hand, professionals who reject physical restraints may perform actions of resistance to minimise their use [42,43]. Finally, it is expected that the way that the professional values the person being cared for will determine the ethical judgment made regarding the application of physical restraint. Accordingly, mechanisation theory may clarify how geriatric patients are viewed and treated, and it may help explain why some professionals legitimise the use of restraints [52].

Finally, this study will identify possible alternatives to physical restraints according to the healthcare professionals consulted and determine whether restraint-free attention is a real possibility in intermediate care centres. However, to do so, the potential barriers to the withdrawal of physical restraints in these healthcare settings and the measures to be adopted in their place must be identified.

### Limitations and Strengths

This research project was impacted by COVID-19. The research was originally designed to incorporate discussion groups as a primary data collection technique, enabling group social constructions to be explored. As a secondary technique, individual interviews were also scheduled with the intention of clarifying aspects that did not emerge in the discussion groups. However, in view of the pandemic situation, legal restrictions on face-to-face meetings, the work overload experienced by healthcare professionals, and the risk of contagion, it was decided to carry out only individual interviews. Furthermore, the present data collection process coincides with the pandemic period. To minimise potential bias from the impact of this new healthcare reality on the use of physical restraints, hospital units that specifically care for COVID-19 patients have been excluded from the study.

This study will provide, from qualitative analysis, the aspects related to healthcare professionals and the institution that determine the use of physical restraints at intermediate care hospitals. The results of this study will lay the foundations to implement a more dignified care model and reduce the use of physical restraints in the care of geriatric patients.

## 4. Conclusions

This study, undertaken from the perspective of the healthcare professionals involved, will contribute to understanding the use of physical restraints in the care of geriatric patients hospitalised in intermediate care units. The findings will be transferred to different audiences so that this use of restraints is no longer a common practice. The population considered in this study, although subject to cognitive deterioration, enjoys full civil rights. Therefore, at the political level, careful consideration should be placed on how these devices are used to determine whether the current healthcare practice is in accordance with the applicable legislation and whether it should be modified to guarantee the quality of care. In addition, the study findings are expected to help managers design programmes for the removal of physical restraint procedures (as appropriate to the health centres concerned) and promote an alternative care model based on patients’ rights and needs. In the educational context, the conclusions drawn from this study will encourage future healthcare professionals during their undergraduate studies to reflect on care systems for geriatric patients and the ethical questions underlying the use of restraints. The study may also make society in general more aware of the ethical conflicts identified and encourage calls for restraint-free care. Finally, with respect to the healthcare system as a whole, the goal pursued in this study is to help reduce or eliminate the use of restraints in the care of geriatric patients.

In summary, it is essential to analyse the healthcare culture, with specific regard to the attitudes and practices of professionals regarding the use of physical restraints in assisting the elderly so that a more dignified care model, free of such restraints, may be proposed and implemented.

## Figures and Tables

**Table 1 ijerph-18-07509-t001:** Profile of participants.

Participants	General Hospital	Joan MarchHospital	SJD Palma Hospital	Total		
				Male	Female	All
Physicians	2–3	2–3	2	4	3	7
Nurses	2–3	2–3	2	2	5	7
Nursing assistants	2–3	2–3	2	1	6	7
Managers	1–2	1–2	1	2	2	4
Total	7–11	7–11	7	9	16	25

**Table 2 ijerph-18-07509-t002:** Inclusion and exclusion criteria.

Inclusion Criteria	
Centres and Units	Intermediate care (medium stay) hospitals Hospitalisation units caring for geriatric patients (>65 years old) with physical and/or cognitive deterioration in medium or long stay situations
Healthcare professionals	Physicians, nurses, and nursing assistants Experience >2 years
Managers	Experience >6 months in a position of responsibility
**Exclusion Criteria**	
Centres and Units	Centres where physical restraints have been completely removed Specific neurorehabilitation or palliative care units Specific units for attending patients with COVID-19

**Table 3 ijerph-18-07509-t003:** Script for the interview with (non-management) healthcare professionals.

Could you comment on the last time you had to physically restrain a patient?
In your opinion, what factors influence the use of physical restraints for a geriatric patient?
María is an 80-year-old woman with mild cognitive impairment who has recently become disoriented. Her daughter has requested assistance because her mother constantly gets up from her chair: What would you do?
What responsibility do you have as a physician/nurse/nursing assistant regarding a physically restrained patient in your unit?
When you are on duty and responsible for a patient who is quite calm, but nevertheless being physically restrained, how would you respond?
(Participant is shown a photo of a previously-physically restrained patient who has fallen out of bed): Do you think this situation could have been prevented?

## Data Availability

The datasets generated and/or analyzed during the current study will not be publicly available due to privacy and confidentiality reasons, but they will be available from the corresponding author upon reasonable request.

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
