# Peer review of "The Use of Physical Restraints on Geriatric Patients: Culture and Attitudes among Healthcare Professionals at Intermediate Care Hospitals in Majorca. A Qualitative Study Protocol"

_ijerph, 2021, doi:10.3390/ijerph18147509_

Round 1

Reviewer 1 Report

Thank you for the opportunity to review this paper. This paper is interesting and important topic to understand attitudes among healthcare professionals towards the physical restraint 18 of geriatric patients in intermediate care hospitals. In addition, the paper aims to understand qualitative study protocol as a method.

Title: the title is clear.

Introduction:

You use concept healthcare professionals, health professionals. Please, harmonise the use of the concept. In addition, concept geriatric patients, elderly persons etc. please, harmonise.

  1. Methods and Analysis:

Number of study participants remains somewhat unclear in the text: “Approximately 10-15 physicians, 45-55 nurses and 55-70 nursing assistants are employed in the hospitals included in the study”. Later in the text: “At least 7-10 healthcare professionals will be recruited from each job category and 3-5 persons in management positions…”

Recruitment process needs more detailed description.

Ethical considerations are described with merit.

Discussion is clear.

Limitations: How about strengths of the study?

Conclusions is clear.

Author Response

Response to Reviewer 1 Comments (Uploaded an attachment):

Thank you for the opportunity to review this paper. This paper is interesting and important topic to understand attitudes among healthcare professionals towards the physical restraint 18 of geriatric patients in intermediate care hospitals. In addition, the paper aims to understand qualitative study protocol as a method.

Thank you very much for your input and your suggestions.

Point 1: You use concept healthcare professionals, health professionals. Please, harmonise the use of the concept. In addition, concept geriatric patients, elderly persons etc. please, harmonise.

Response 1: We have harmonized these concepts, using ‘healthcare professionals’ and ‘geriatic patients’ ones.

Point 2: Number of study participants remains somewhat unclear in the text: “Approximately 10-15 physicians, 45-55 nurses and 55-70 nursing assistants are employed in the hospitals included in the study”. Later in the text: “At least 7-10 healthcare professionals will be recruited from each job category and 3-5 persons in management positions…”.

Response 2: The first sentence makes a reference to the study population: the number of healthcare professionals employed in each of the hospitals included in the study. The second one alludes to the number of study participants: this is the minimum number of professionals from each job category that will be recruited, although as many professionals as necessary will be interviewed to achieve saturation of the data. We have clarified it in the main manuscript.

Point 3: Recruitment process needs more detailed description.

Response 3: We have detailed description of the recruitment process, including the profile of the collaborators, the information provided to the professionals who expressed interest in participating in the study, and the snowball strategy.

Point 4: Limitations: How about strengths of the study?

Response 4: We have included the strengths of the study in the limitations section and changed the title section: limitations and strengths.

Reviewer 2 Report

Thank you for having the opportunity to rewiev this very interesting paper. Physical restraints are still frequently used in the management of patients, older people in particular. Recently, the negative outcomes of physical restraint use have often been reported, but very limited research effort has been made to examine whether such nursing practice have any adverse effects on patients' quality of life, self-reliance, autonomy and dignity, stigmatisation, mood, anxiety or length of stay  in hospitals.

My suggestions:

Introduction. In my opinion, the use of physical restraints on older patients gives rise to a number of ethical issues. It has been suggested e.g. that physical restraints hinder the promotion of self-reliance in older people and disrespect older people's autonomy and dignity. I think this point of view should be explained in the Introduction part.

Introduction. You write that in Spain, htere is no national normative or legislative framework regarding this question - and how is the situation in other European countries? Is there lieterature about? Please include this information.

Methods and Analysis. Ethnomethodology seeks to understand the method by which individuals construct, negotiate, and agree upon reality, but questions the possibility of an objective science of the subjective human condition - in the part Methods and analysis, in my opinion, this term is explained to short.

Methods and Analysis. You include as participants physicians, nurses and nursing assistants. And what is with other medical proffesionals - e.g. physiotherapiests? Please explain.

Limitations. I think You should also include the interviewer-respondent interaction and write about the potential interview bias.

I whish all the best to all coarthors. Great idea and well written article.

Author Response

Response to Reviewer 2 Comments (Uploaded an attachment):

Thank you for having the opportunity to rewiev this very interesting paper. Physical restraints are still frequently used in the management of patients, older people in particular. Recently, the negative outcomes of physical restraint use have often been reported, but very limited research effort has been made to examine whether such nursing practice have any adverse effects on patients' quality of life, self-reliance, autonomy and dignity, stigmatisation, mood, anxiety or length of stay  in hospitals.

Thank you very much for your input, which we believe will result in the improvement of the manuscript. Please find below the answers to the points raised.

Point 1: Introduction. In my opinion, the use of physical restraints on older patients gives rise to a number of ethical issues. It has been suggested e.g. that physical restraints hinder the promotion of self-reliance in older people and disrespect older people's autonomy and dignity. I think this point of view should be explained in the Introduction part.

Response 1: In the introduction part we have expanded the ethical discussion on the use of physical restraints in the paragraph about the reasons why the use of restraints is currently being questioned.

Point 2: You write that in Spain, htere is no national normative or legislative framework regarding this question - and how is the situation in other European countries? Is there lieterature about? Please include this information.

Response 2: There is quite literature about specific European countries legislative framework. We have added information about some international documents that must be considered and make reference to some countries that have legislated the use of physical restraints.

Point 3: Ethnomethodology seeks to understand the method by which individuals construct, negotiate, and agree upon reality, but questions the possibility of an objective science of the subjective human condition - in the part Methods and analysis, in my opinion, this term is explained to short.

Response 3: We have explained with more detail some characteristics of ethnomethodology.

Point 4: Methods and Analysis. You include as participants physicians, nurses and nursing assistants. And what is with other medical proffesionals - e.g. physiotherapiests? Please explain.

Response 4: We have included physicians, nurses and nursing assistants because they are those identified by literature as the main ones involved in decision making, prescription and application of physical restraints. These professionals are those implied in continued care in hospitalization units, where physical restraints are usually applied. Other professionals like physiotherapist, occupational therapist and social worker have a fundamental role in rehabilitation process but they are not present 24h at the units, nor during the night that’s when decision-making about physical restraints takes place. In case the participants of the study identify that other healthcare professionals have a fundamental role in the use of physical restraints, we will interview them. We have clarified it in the main manuscript.

Point 5: Limitations. I think You should also include the interviewer-respondent interaction and write about the potential interview bias.

Response 5: To ensure that the participant's discourse can emerge freely, the main researcher has carried out a process of reflexivity in which she has questioned herself about her position on the use physical restraints. In this sense, the process of reflexivity has been carried out previously and during the research. This appreciation has been more explained in the Scientific Rigour part, when explaining process of reflexivity as a rigour technique.
